# Intramuscular Neural Distribution of the Gluteus Maximus Muscle: Diagnostic Electromyography and Injective Treatments

**DOI:** 10.3390/diagnostics14020140

**Published:** 2024-01-08

**Authors:** Kyu-Ho Yi, Dong Chan Kim, Siyun Lee, Hyung-Jin Lee, Ji-Hyun Lee

**Affiliations:** 1Division in Anatomy and Developmental Biology, Department of Oral Biology, Human Identification Research Institute, BK21 PLUS Project, Yonsei University College of Dentistry, Seoul 03722, Republic of Korea; kyuho90@daum.net; 2Maylin Clinic (Apgujeong), Seoul 06005, Republic of Korea; 3Department of Rehabilitation Medicine, Eunpyeong St. Mary’s Hospital, Seoul 03312, Republic of Korea; dckim94@gmail.com; 4Department of Cancer Biology, Dana-Farber Cancer Institute, Harvard Medical School, Boston, MA 02215, USA; siyun@broadinstitute.org; 5Catholic Institute for Applied Anatomy, Department of Anatomy, College of Medicine, The Catholic University of Korea, Seoul 06591, Republic of Korea; 6Department of Anatomy and Acupoint, College of Korean Medicine, Gachon University, Seongnam 13120, Republic of Korea

**Keywords:** EMG, botulinum toxin, spasticity, gluteus maximus muscle, Sihler staining, intramuscular neural distribution

## Abstract

Introduction: The purpose of this study was to investigate neural patterns within the gluteus maximus (Gmax) muscle to identify optimal EMG placement and injection sites for botulinum toxin and other injectable agents. Methods: This study used 10 fixed and 1 non-fixed adult Korean cadavers. Intramuscular arborization patterns were confirmed in the cranial, middle, and caudal segments of 20 Gmax muscles using Sihler staining. Ultrasound images were obtained from one cadaver, and blue dye was injected using ultrasound guidance to confirm the results. Results: The intramuscular innervation pattern of the Gmax was mostly in the middle part of this muscle. The nerve endings of the Gmax are mainly located in the 40–70% range in the cranial segment, the 30–60% range in the middle segment, and the 40–70% range in the caudal segment. Discussion: Addressing the spasticity of the gluteus maximus requires precise, site-specific botulinum toxin injections. The use of EMG and other injection therapies should be guided by the findings of this study. We propose that these specific sites, which correspond to areas with the densest nerve branches, are the safest and most efficient locations for both botulinum toxin injections and EMG procedures.

## 1. Introduction

Spasticity, which commonly results from conditions that affect the central nervous system, such as stroke, cerebral palsy, traumatic brain or spinal cord injury, and multiple sclerosis, significantly contributes to functional impairment in affected individuals [1].

The gluteus maximus (Gmax) muscle is essential in extending the thigh, stabilizing the leg during walking, and standing from a stooped position. It is involved in outward thigh turning, is continuously active during the strong lateral rotation of the thigh, and is intermittently active during walking and climbing stairs [2]. The Gmax muscle is often affected by long-lasting spasticity, and abnormalities from excessive activity can significantly impair functions such as gait [3]. Currently, injections of botulinum neurotoxin (BoNT) are considered one of the safest and most effective treatments for spasticity in the Gmax [4,5,6,7,8]. Considering that the effects of BoNT depend on the dosage, it is crucial to administer an appropriate amount of BoNT to ensure adequate levels of neurotoxin at the neuromuscular junctions of the muscle [9,10,11,12].

BoNT is also becoming increasingly important in the treatment of myofascial pain syndrome (MPS), a complex, chronic pain disorder primarily characterized by the presence of myofascial trigger points in muscles [13]. These trigger points are sensitive areas within the muscle or fascia that cause pain and can lead to muscle stiffness, decreased range of motion, and referred pain. MPS is often associated with muscle overuse or stress [14]. Injections of local anesthetics, steroids, normal saline, and BoNT are frequently administered as part of the treatment [13,15].

When treating spasticity and MPS, it is crucial to accurately identify the precise location of pain within the muscle tissue by understanding the intramuscular neural distribution pattern [16,17,18]. The clinical significance of targeting injections to the neural arborized areas, where most neuromuscular junctions are located, has been established through research studies conducted on other muscles [19,20].

To minimize potential side effects and enhance the effectiveness of BoNT and other injective treatments, it is essential to locate the medications as close as possible to areas with neural arborization. Precise injection into these areas is crucial for maximizing the clinical efficacy of BoNT injections, and it has been demonstrated to significantly reduce the volume required for intramuscular efficacy [17,20]. Therefore, extensive research has identified the anatomical locations of neuromuscular junctional areas in various muscles to facilitate accurate targeting during injection [12,21,22,23].

On a related note, electromyography (EMG) plays a crucial role in the diagnosis of various neuromuscular disorders, and it is used extensively to evaluate muscle activity by measuring electrical signals [24]. In previous research, EMG studies have often treated the Gmax muscle as a unified entity. This approach regrettably overlooks the intricate segmentation of the Gmax into three distinct segments: cranial, middle, and caudal. Each of these segments plays a unique functional role, so each should be analyzed individually to enable a comprehensive understanding of the muscle functions [25,26,27].

The cranial segment of the Gmax, likely rich in type 1 (slow-twitch) muscle fibers, plays a crucial role in postural activities, particularly hip adduction and flexion during locomotion. Conversely, the middle and caudal segments, characterized by a higher proportion of type 2 (fast-twitch) muscle fibers, are essential for more dynamic tasks, including the powerful extension and external rotation of the hip [26]. To accurately define these segments, the Gmax can be divided using a normalized line connecting the iliac crest, the posterior superior iliac spine (PSIS), the sacrum, and the coccyx (CO) and considering the muscle width from origin to insertion. A line extending from the greater trochanter (GT) to the lateral border of the femur helps to define the cranial and caudal borders of the muscle insertion points. In that way, the Gmax is effectively segmented into three equal portions with distinct roles and contractile properties [26]. This nuanced segmentation not only aligns with the anatomical structure of Gmax, but also emphasizes its diverse functional capabilities, underscoring the importance of segment-specific analyses in EMG studies for targeted muscle training and rehabilitation.

Analyzing the muscle activity in great detail allows clinicians and researchers to tailor rehabilitation regimens with good precision, maximizing the potential for successful recovery and satisfactory patient outcomes [5]. Such a detailed analysis becomes particularly critical when conducting separate EMG investigations on each of the three segments of the Gmax. However, until relatively recently, clinical understanding of the Gmax function was oversimplified. A previous study investigating the accuracy of fine-wire electrode placement in EMG procedures revealed that unintentional placement in muscles other than the intended segments significantly reduced the accuracy of muscle signal sampling [28].

Moreover, EMG combined with a knowledge of intramuscular neural distribution can play a valuable role in diagnosing neural injuries by providing a comprehensive assessment of the condition. This integrated approach enhances the precision and effectiveness of therapeutic interventions for MPS.

Using conventional gross anatomical dissection methods to identify the neural distribution within muscles and to determine the precise locations for BoNT injections and electrode placement has been challenging. These methods have limitations, such as the potential risk of nerve damage during dissection process. Furthermore, no previous research has segmented the Gmax into three parts to examine the patterns of neural distribution within each segment.

Therefore, we used the Sihler’s whole nerve staining technique, a method that renders muscles transparent while staining nerves, to effectively display the neural distribution within muscles without harming the nerves [29]. Using this technique, we analyzed the neural distribution patterns in each segment of the Gmax. Based on those findings, we propose appropriate locations for BoNT injections and electrode placement.

## 2. Materials and Methods

We dissected 20 Gmax muscles obtained from 10 embalmed Korean cadavers (5 males and 5 females, mean age 81.7 years; range 74–87 years) and subjected them to Sihler staining to identify the distribution of intramuscular nerve endings. Additionally, to validate the practical applicability of our research findings, we conducted ultrasonography on a non-embalmed cadaver and attempted ultrasound-guided injections.

### 2.1. Modified Sihler Staining Procedure

The skin of each embalmed cadaver was carefully removed, and the subcutaneous tissue was cleared to expose the Gmax muscles. Then, the muscles were procured to obtain a visual representation of the intramuscular neural distribution. Sihler staining involves several critical stages, each of which contributes to the comprehensive visualization of intramuscular neural distribution [18,29]. The modified Sihler staining procedure that we used, adapted for the specific characteristics of the Gmax, is outlined below. All reagents were purchased from Daejung Chemicals & Metals, Siheung-si, Gyeonggi-do, Republic of Korea.

Fixation: The Gmax specimens were placed in a 10% formalin solution (for one week for fixation).Maceration and Depigmentation: After fixation, the specimens were soaked in running tap water for one hour and then submerged in a 3% aqueous potassium hydroxide and 0.01% hydrogen peroxide solution for 3 to 4 weeks for maceration and depigmentation. We monitored the solution closely and replaced it whenever it became cloudy, which is essential for subsequent staining. Because of the substantial thickness of the Gmax muscle belly, we applied manual manipulation such as gentle kneading to ensure effective maceration.Decalcification: For whitening and decalcification, the specimens were soaked in Sihler 1 solution (10% glycerin and 7% glacial acetic acid) for a week.Staining: The specimens were soaked in Sihler 2 solution (10% glycerin and 10% Ehrlich’s hematoxylin) for one week to stain the neural structures.Destaining: The stained muscle specimens were re-immersed in Sihler 1 solution to remove excess staining. This step ensures that only neural structures retain the stain for clear visualization.Neutralization: After destaining, the specimens were rinsed in flowing water for 40 min and then soaked in 0.05% lithium carbonate solution for 50 min to intensify the nerve color.Clearing: In the final stage, the neutralized Gmax specimens were gradually brought to transparency using a cleaning solution of 1:1 glycerin and formamide. The concentration of the solution was increased from 70% to 100% in 10% increments as the solution was changed daily for several days. This process makes the muscle tissue transparent and emphasizes the stained nerves (Figure 1).Identification and Analysis of Neural Distribution Patterns: After Sihler staining was complete, each stained Gmax was placed on a vinyl-wrapped LED view box. The distribution of nerves was observed using a magnifying glass. Each Gmax was divided into three segments, caudal, middle, and cranial, each of which was further divided into ten equal parts based on the width from the muscle origin to its insertion. The presence and distribution of nerve endings in each part were examined. Then, the proportion of each area containing nerve endings was calculated to determine the intramuscular neural distribution ratio across the entire muscle.

### 2.2. Ultrasonography and Ultrasound-Guided Injections

Prior to the ultrasonographic examination, anatomical landmarks were used to divide the Gmax into three segments. The PSIS–GT line, connecting the PSIS to the GT, indicates the cranial segment of the Gmax. The CO–IT line, linking the CO to the ischial tuberosity (IT), marks the caudal segment. The middle line, intersecting the midpoint between the PSIS–GT and CO–IT lines, divides the muscle’s sagittal plane. Ultrasonographic images were obtained following those lines (Figure 2A).

After establishing those landmarks, the PSIS–GT line was divided into thirds, and perpendicular oblique lines were drawn at its one-third and two-thirds points. Those lines were used to obtain horizontal transverse-sectional ultrasonographic images of the Gmax (Figure 2B).

The injection sites for each segment of the Gmax were determined as follows (Figure 2C). Injections were administered along a perpendicular line intersecting the proximal one-third of the PSIS–GT line. More specifically, they were administered at the upper one-third of the PSIS–GT line (1), at the middle line (2), and at the lower one-third between the middle line and the CO–IT line (3). To ensure precise injection within the Gmax, the depth was monitored using ultrasonography to track the injection needle (Figure 2D). A long 22-gauge needle with a 3 mL syringe was used to inject blue dye at those points. After the injections, the skin and subcutaneous tissue of the cadaver were removed to expose the Gmax and confirm the dye injection sites.

## 3. Results

### 3.1. Neural Distribution Patterns in the Gmax

Across the examined specimens, we observed that intramuscular neural distributions within the muscle belly of the Gmax were predominantly concentrated in the central regions (Figure 3).

### 3.2. Cranial Segment

In the cranial segment of the Gmax, nerve endings were identified in every sample from the 20% to 70% sections. The occurrence of nerves was less frequent in the most proximal (0–10%) and most distal (90–100%) sections (Table 1).

### 3.3. Middle Segment

Nerve endings were present in all samples from the 40% to 70% sections in the medial segment of the Gmax. Nerves were less abundant in the sections below 20% and above 70% (Table 1).

### 3.4. Caudal Segment

In the caudal region, all samples within the 30% to 70% sections exhibited nerve endings. The occurrence decreased in the sections below 20% and above 80% (Table 1).

### 3.5. Ultrasonography and Ultrasound-Guided Injections

Using the landmarks established, we were able to easily and accurately obtain vertical ultrasonographic images of each segment of the Gmax (Figure 4A–C), as well as transverse images (Figure 4D,E), through the virtual reference lines. Furthermore, we confirmed that the ultrasound-guided injections using these reference lines were delivered precisely to the targeted central areas of each of the three segments of the Gmax (Figure 5).

## 4. Discussion

This study used Sihler’s whole nerve staining technique to reveal the distribution pattern of intramuscular neural arborization in the Gmax by segment. We confirmed that the nerve endings of the Gmax are mainly located in the 40–70% range in the cranial segment, the 30–60% range in the middle segment, and the 40–70% range in the caudal segment.

Only one previous paper confirmed the intramuscular neural distribution pattern in the Gmax. In that study, Junxi Wu et al. also visualized the intramuscular neural distribution of Gmax through Sihler’s staining, but they based their analysis on the surface structure of the ilium crest and gluteal sulcus [30], so they could not confirm the distribution pattern in the three segments of the Gmax. We first classified the three segments and then identified the nerve distribution pattern within each one.

Our findings in this study are important for improving the placement accuracy of EMG electrodes and BoNT injections, and they could have important implications for treating conditions such as spasticity and MPS.

EMG is a diagnostic method that assesses and records the electrical activity generated by skeletal muscles when electrically or neurologically activated [31]. These signals are complex and controlled by the nervous system according to the anatomical and physiological characteristics of the muscles. EMG is a key tool in medical and research settings for assessing muscle health and function [31]. However, accurate EMG results require the precise placement of electrodes on specific muscles and in specific areas [32,33].

BoNT is a safe and effective treatment for focal spasticity [34]. Its primary therapeutic effect is to inhibit muscle contraction at the neuromuscular junction and break the cycle of pain [34,35,36,37]. The main side effect of injection treatments, including BoNT, is the development of unwanted muscle tissue, along with symptoms such as weakness, infection, pain, and hematoma [35,38]. Furthermore, if a significant amount of BoNT is injected repeatedly, the body can produce antibodies that reduce the effectiveness of the treatment [12,15,21]. Therefore, to maximize treatment effectiveness and minimize side effects, an appropriate amount of BoNT must be injected directly into the neuromuscular junction; that is, the nerve ending area.

Currently, no standard injection site for BoNT therapy at the Gmax has been established. It is important to administer an appropriate amount of BoNT so that the toxin can sufficiently penetrate the nerve-segmented area. In this study, we used Sihler staining to obtain an accurate and comprehensive understanding of intramuscular neural distribution and overcome the limitations of manual dissection. According to previous research results, BoNT injections properly targeted to the nerve ending distribution area resulted in a noticeable decrease in muscle volume compared with the existing method [19]. The detailed map of intramuscular innervation provided in this study could be used to establish a standardized and targetable method for placing both EMG electrodes and injection therapies such as BoNT, maximizing their efficacy. Such precision is also expected to minimize the risk of side effects.

Ultrasound is used for muscle detection, especially when dealing with muscles deep inside the body. It is desirable to use ultrasound imaging when performing injection treatments and EMG electrode placement. With ultrasound, subcutaneous fat, muscles, blood vessels, nerves, etc., can be observed in real time, so it is used to prevent iatrogenic complications such as nerve trunk or blood vessel damage [39,40]. In this study, we chose three surface landmarks for easy ultrasound examination and performed injections under ultrasound guidance to safely verify whether the injection was administered accurately to the analyzed area.

Although we used only one cadaver for this test, the injection was confirmed to be successful. Therefore, we suggest that clinicians perform ultrasound-assisted injections rather than blind injections.

This study has the following limitations. It relies on a relatively small sample of 20 Gmax muscles from 10 cadavers, which might limit the comprehensive understanding of innervation patterns. Additionally, the use of cadaveric tissue, while insightful, might not fully represent the dynamic properties of living human muscle. Additionally, the limited age range (65–79 years) of the cadaveric specimens might not reflect potential changes in neural patterns caused by factors such as age, health, and muscle activity.

Additionally, individual anatomical variations in the Gmax could not be detected in this research and were not considered. Moreover, verification using ultrasonic waves has the fatal disadvantage of being difficult to propose as an appropriate method because it was used in only one case. Therefore, the future verification of our results is needed through additional cadaveric studies or clinical studies on living patients.

## 5. Conclusions

In this study, we used Sihler’s staining of Gmax muscles from human cadavers to reveal intramuscular neural arborization patterns. Our findings will be helpful in diagnostic EMG and injection therapies such as BoNT, which will have important implications for the clinical management of conditions such as spasticity and MPS.

Our findings show that nerve endings in the Gmax are mainly located in the central region. This knowledge will help to guide BoNT injections and EMG electrode placement, improving treatment efficacy and safety while reducing side effects. Targeting injections to these nerve-dense areas is expected to optimize treatment and maximize clinical benefit.

## Figures and Tables

**Figure 1 diagnostics-14-00140-f001:**
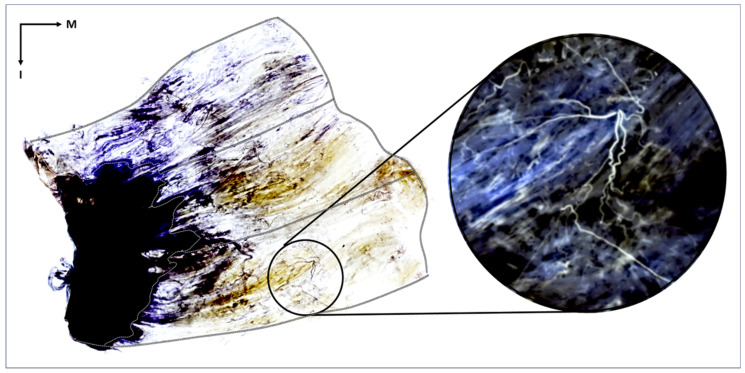
The stained gluteus maximus muscle reveals the distribution of intramuscular nerves. The tendinous portion of the gluteus maximus is the black region outlined with a white line, and the outer border of the muscle is delineated with a black line. The muscle fibers appear transparent, and the nerve fibers are stained violet. The distribution of intramuscular nerves can be seen within the muscle belly. In the right magnified image, the nerve twigs are shown within the inferior-middle part of the muscle. The image in the circle has been artificially inverted so that the nerves are white to better highlight the intramuscular neural distribution. I, inferior; M, medial.

**Figure 2 diagnostics-14-00140-f002:**
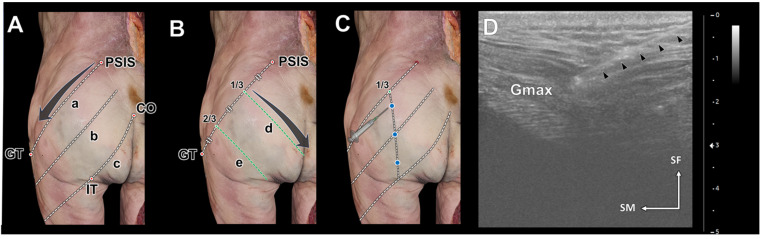
The reference lines for the ultrasonography scanning procedure (**A**,**B**) and ultrasound-guided injection procedure (**C**,**D**). Ultrasonographic scanning of the gluteus maximus (Gmax) in the longitudinal orientation was performed along three reference lines: the posterior superior iliac spine (PSIS)–greater trochanter of the femur (GT) line (a), the middle line (b), and the coccyx (CO)–ischial tuberosity (IT) line (c). Ultrasonographic scanning in the transverse orientation of the Gmax was conducted along three perpendicular lines (d and e) arranged on the PSIS–GT line at equal intervals. The injection points were set at three points along a vertical line from the 1/3 point of the PSIS–GT line (blue dots in (**C**)). The positions of the Gmax and the needle were monitored continuously during the procedures (**D**). The needle tip was safely positioned within the Gmax, as indicated by a black arrowhead. SF, superficial; SM superomedial.

**Figure 3 diagnostics-14-00140-f003:**
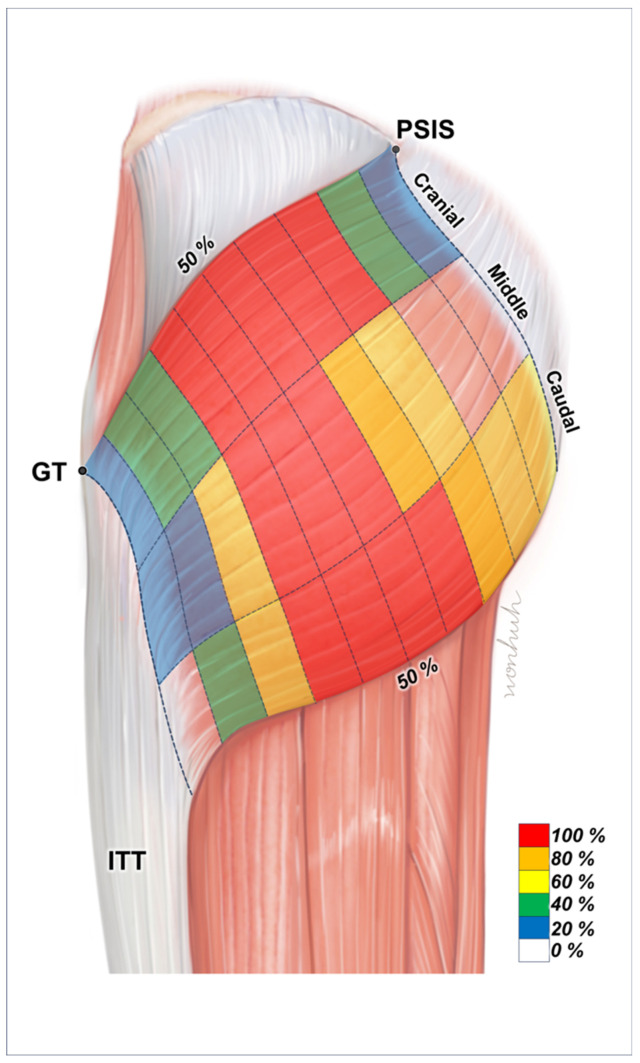
Illustration of the intramuscular innervation patterns of the gluteus maximus (Gmax). The nerve endings are primarily located in the central regions of the muscle. The areas shaded in red indicate ideal points for injecting botulinum neurotoxin into the Gmax. PSIS, posterior superior iliac spine; GT, greater trochanter of the femur; ITT, iliotibial tract.

**Figure 4 diagnostics-14-00140-f004:**
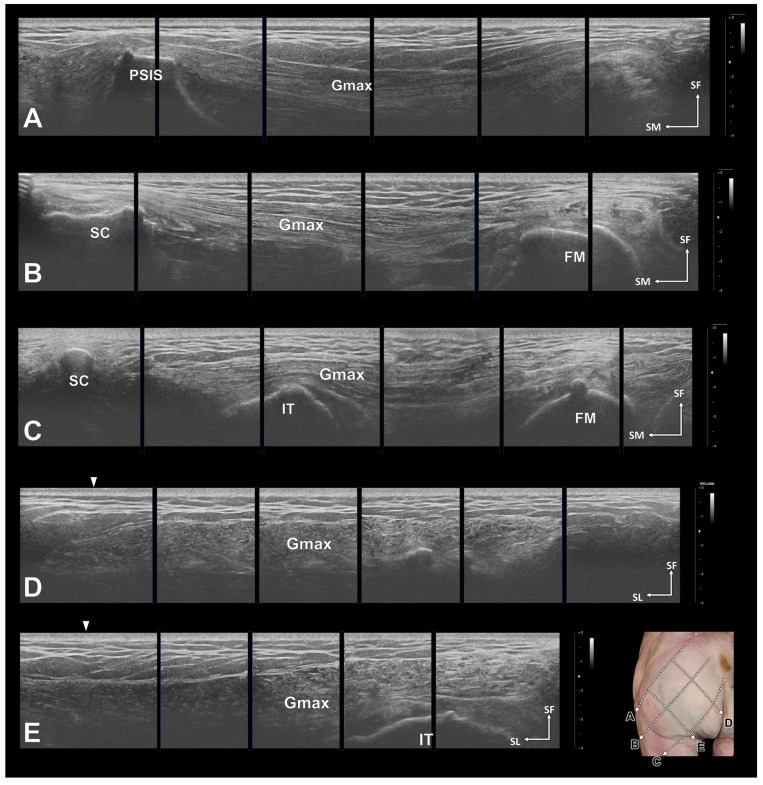
Ultrasonographic images of the gluteus maximus (Gmax). In all images, the morphology of the Gmax is clear. Longitudinal sectional images of the Gmax along the posterior superior iliac spine (PSIS)–greater trochanter of the femur (GT) line (**A**), the middle line (**B**), and the coccyx (CO)–ischial tuberosity (IT) line (**C**). Transverse sections were obtained along a vertical line from the proximal 1/one-third of the PSIS–GT line (**D**) and from the two-thirds point (**E**). The white arrows in D and E indicate the PSIS–GT line. FM, femur; SC, sacrum; SF, superficial; SL, superolateral.

**Figure 5 diagnostics-14-00140-f005:**
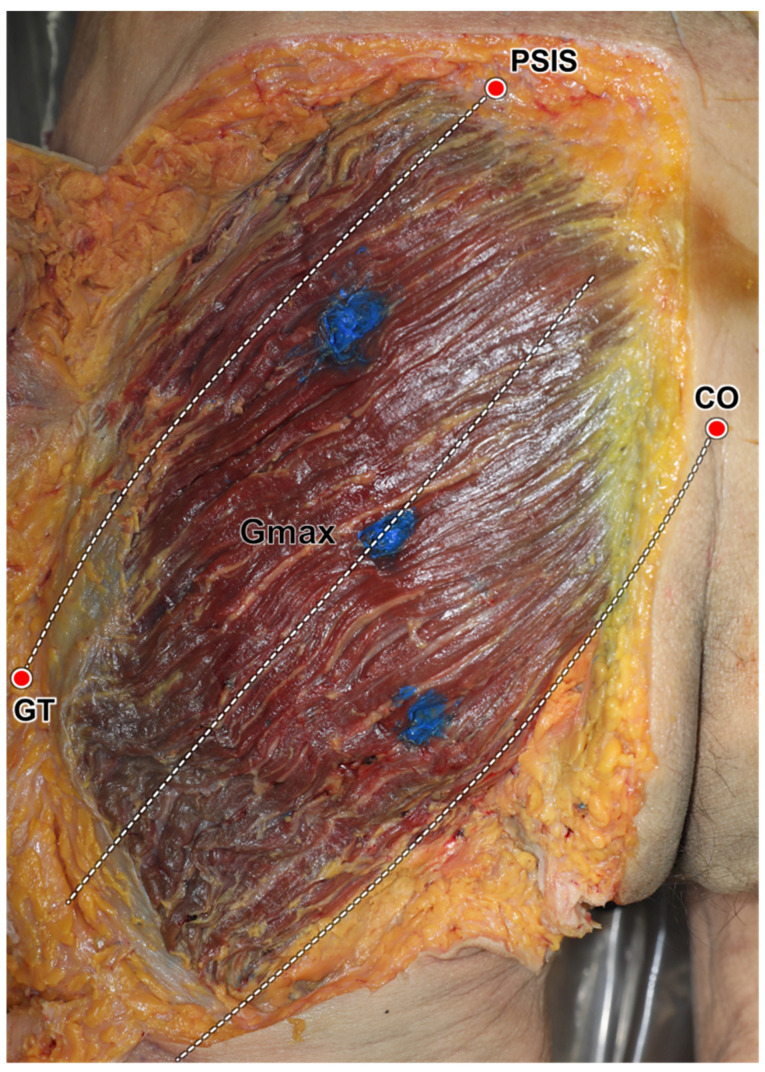
Verification of ultrasound-guided injections using the reference lines. Based on the suggested anatomical landmarks, the dye was injected into the three segments of the gluteus maximus (Gmax). PSIS, posterior inferior ischial spine; GT, greater trochanter of femur, CO, coccyx.

**Table 1 diagnostics-14-00140-t001:** The distribution area of the inferior gluteal nerve to the three parts of the gluteus maximus muscle.

Proportion within the Gluteus Maximus	Cranial Segment	Middle Segment	Caudal Segment
0–10%	4/20 (20%)	0/20 (0%)	4/20 (20%)
10–20%	8/20 (40%)	0/20 (0%)	4/20 (20%)
20–30%	20/20 (100%)	12/20 (60%)	16/20 (80%)
30–40%	20/20 (100%)	16/20 (80%)	20/20 (100%)
40–50%	20/20 (100%)	20/20 (100%)	20/20 (100%)
50–60%	20/20 (100%)	20/20 (100%)	20/20 (100%)
60–70%	20/20 (100%)	20/20 (100%)	20/20 (100%)
70–80%	8/20 (40%)	12/20 (60%)	12/20 (60%)
80–90%	8/20 (40%)	4/20 (20%)	8/20 (40%)
90–100%	4/20 (20%)	4/20 (20%)	0/20 (0%)

## Data Availability

Upon request to corresponding author.

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
