# Peer review of "Intramuscular Neural Distribution of the Gluteus Maximus Muscle: Diagnostic Electromyography and Injective Treatments"

_diagnostics, 2024, doi:10.3390/diagnostics14020140_

Round 1
Reviewer 1 Report
Comments and Suggestions for Authors
The presented study aims to identify optimal electromyography placement and injection sites within the gluteus maximus muscles by mapping out nerve patterns. This is achieved using a modified Sihler method, and the findings are instrumental in suggesting effective and secure locations for botulinum toxin injections and electromyography procedures.
1. Figure Uniformity: It might be beneficial to standardize the format, including font style and figure layouts, to ensure a coherent presentation throughout the paper.
2. Figure 3: Please review the "gluteus maximus" labeling for accuracy.
3. Correlation Clarity: The paper suggests a connection between Myofascial Trigger Points (MTrPs) and nerve entry points, but deeper analysis or clear conclusions are lacking.
4. Muscle Variation: Were any variations in the gluteus maximus muscle anatomy observed?
Author Response
Response to Reviewer 1
We are so pleased to have your precious time with constructive comments. Thank you for noticing the error, we have changed the content of the manuscript as per your suggestion.
Comment 1. Figure Uniformity: It might be beneficial to standardize the format, including font style and figure layouts, to ensure a coherent presentation throughout the paper.
Answer 1: We have replaced all figures and unified the format.
Comment 2. Figure 3: Please review the "gluteus maximus" labeling for accuracy.
Answer 2: The figure has been replaced, and we have considered the specified content in our revision.
Comment 3. Correlation Clarity: The paper suggests a connection between Myofascial Trigger Points (MTrPs) and nerve entry points, but deeper analysis or clear conclusions are lacking.
Answer 3: We sought to explain the relationship by proposing treatment methods for MPS
1) When treating spasticity and MPS, it is crucial to accurately identify the precise location of pain within the muscle tissue by understanding the intramuscular neural distribution pattern
2) Our findings in this study are important for improving the placement accuracy of EMG electrodes and BoNT injections, and they could have important implications for treating conditions such as spasticity and MPS.
3) This integrated approach enhances the precision and effectiveness of therapeutic interventions for MPS.
Comment 4. Muscle Variation: Were any variations in the gluteus maximus muscle anatomy observed?
Answer 4: We did not observe any notable variations in the Gmax in any of the cadavers.
Reviewer 2 Report
Comments and Suggestions for Authors
Please see the attached PDF with comments and suggestions on the manuscript to be considered by the authors.

Author Response
Response to Reviewer 2
We are so pleased to have your precious time with constructive comments. Thank you for noticing the error, we have changed the content of the manuscript as per your suggestion.
Comment 1- Lines 23 and 24: the authors should clarify what exactly “posterior iliac crest” means.
Answer: Deeply grateful for your kind remark, we have completely revised the abstract and removed the mentioned phrase.
Comment 2- Lines 41-43: the sentence should be rewritten in order to better describe the actions of the gluteus maximus muscle (please see, for instance, Gray’s Anatomy. The Anatomical Basis of Clinical Practice. Elsevier, 42nd ed., pag. 1377).
Answer: Deeply appreciative of your guidance, we have rewritten the sentence as follows, drawing on the content of the suggested reference.
The gluteus maximus (Gmax) muscle is essential in extending the thigh, stabilizing the leg during walking, and standing from a stooped position. It is involved in outward thigh turning, is continuously active during strong lateral rotation of the thigh, and intermittently is active during walking and climbing stairs [2].
Comment 3- Line 49: “…in the gluteus muscle…” should be replaced by “…in the gluteus maximus muscle…”.
Answer: The content has been revised.
Comment 4- Lines 60-62: the sentence should be rewritten in order to better describe the actions of each mentioned segment of the gluteus maximus muscle, and reference(s) should be added.
Answer: The sentence has been modified as follows, and references have been added.
This approach regrettably overlooks the intricate segmentation of the Gmax into three distinct segments: cranial, middle, and caudal. Each of these segments plays a unique functional role, so each should be analyzed individually to enable a comprehensive understanding of the muscle functions [25-27].
Comment 5- Lines 70-73: The “…illuminating study…” must be indicated.
Answer: We have modified the content follows as, and accordingly and added reference.
A previous study investigating the accuracy of fine-wire electrode placement in EMG procedures revealed that unintentional placement in muscles other than the intended segments significantly reduced the accuracy of muscle signal sampling [28].
Comment 6- Lines 112 and 113: the sentence should be rewritten because it repeats information previously mentioned (lines 108 and 109).
Answer: We have deleted the repetitive phrase.
Comment 7- Lines 178 and 179: the legend of Figure 1 should be rewritten because we can see only part of the origin of the gluteus maximus muscle as well as part of its distal attachment.
Answer: We decided that this figure was unnecessary and have removed it.
Comment 8- In all Figures, the authors should include a scheme that indicates the orientation (medial, lateral, superior, inferior).
Answer: "We have changed all the figures, and specifically for Figure 3, we have illustrated the three segments of the Gmax.
Comment 9- Lines 185-187: in the Reviewer opinion this sentence is not clear. Please rewrite.
Answer: We have revised the sentence as follows;
In the cranial segment of the Gmax, nerve endings were identified in every sample from the 20% to 70% sections. The occurrence of nerves was less frequent in the most proximal (0–10%) and most distal (90–100%) sections (Table 1).
Comment 10- Lines 200 and 201: in the legend of Figure 3 “…gluteus muscle…” should be replaced by “…gluteus maximus muscle…”.
Answer: We have removed this figure and replaced it.
Comment 11- Lines 211-222: references should be included in this paragraph.
Comment 12- Lines 212 and 213: the authors should clarify what exactly “inner upper ilium” means.
Comment 13- Lines 214 and 215: the “aponeurosis of the erector spinae” and the “lumbodorsal fascia” are not the same anatomical structure (please see, for instance, Gray’s Anatomy. The Anatomical Basis of
Clinical Practice. Elsevier, 42nd ed., pags. 814 and 845/846).
Comment 14- Lines 221-222: “…the gluteal tuberosity of the linea aspera, between the vastus lateralis and adductor magnus. If present, the third trochanter also serves as an attachment.” This part of the paragraph should be corrected (please see, for instance, Gray’s Anatomy. The Anatomical Basis of Clinical Practice. Elsevier, 42nd ed., pags. 1362 and 1363: descriptions of the gluteal tuberosity and linea aspera).
Answer 11~14: We deemed this passage unnecessary and have removed it, as some of its content was repetitive in the Introduction.
Comment 15- Lines 231-246: references should be included in this paragraph.
Answer: We decided that content was unnecessary and have removed it.
Comment 16- Lines 252-254: the reference number 35 is not from Botwin and colleagues. Please confirm.
Answer: We decided that content was unnecessary and have removed it.
Comment 17- Lines 300 and 301: in the Reviewer opinion, the data of the ultrasonographic image of the gluteus maximus muscle should be indicated (for instance, male or female, age and side).
Answer: We have removed this figure and replaced it with another, incorporating information about the participants in response to the reviewer's feedback